# A Literature Database Review of the Competitive Factors That Influence the Production and Use of Whey in the Brazilian Dairy Industry

**DOI:** 10.3390/foods12183348

**Published:** 2023-09-07

**Authors:** Jamile Schaefer, Bianca Inês Etges, Jones Luís Schaefer

**Affiliations:** 1Health Sciences Department, University of Santa Cruz do Sul (UNISC), Santa Cruz do Sul 96815-900, Brazil; jamileschaefer@gmail.com (J.S.); bianca@unisc.br (B.I.E.); 2Industrial and Systems Engineering Graduate Program, Pontifical Catholic University of Parana (PUCPR), Curitiba 80215-901, Brazil

**Keywords:** whey, milk, competitive factors, dairy products

## Abstract

The consumption of milk and derivatives is part of the diet of a large part of the population. The substantial demand for dairy products has prompted the industry to expand its product range by incorporating whey, a previously disregarded by-product, as a significant ingredient. Consequently, the application of whey within the dairy sector has escalated, fostering novel food trends driven by market, nutritional, technical, environmental, and economic considerations. In this context, the primary objective of this research was to identify, categorise, and analyse the competitive factors influencing the production and use of whey within the dairy products industry and to correlate these factors with Brazil’s economic, food, and nutritional landscape. A comprehensive literature review encompassing 41 articles and scientific documents sourced from PubMed, Scopus, and SciELO databases and supplementary research was undertaken to pinpoint these factors. A total of seventeen competitive factors that exert influence over the production and use of whey within the dairy industry were identified. These seventeen factors were systematically classified and modelled into a hierarchical decision tree structure. A meticulous analysis of these factors revealed a spectrum of strengths, weaknesses, opportunities, and threats associated with this sector, considering the Brazilian nutritional, food, and economic context. This research will help elaborate healthy strategies for developing new products with whey in the composition and maintaining the nutritional quality for the consumer. In addition, the research can help companies manage their operations, which can be directed towards improving their performance in the factors discussed in this research, with reflections and impacts on competitiveness in nutritional, environmental, economic, technological, and organisational dimensions.

## 1. Introduction

Milk is a complex substance that contains high concentrations of macro- and micronutrients [1]. Its versatility makes it possible to consume it in various ways, whether in its natural state or in the form of processed derivative products [2]. The daily consumption of milk and dairy products is a part of diets and nutritional guidelines for health promotion [3], a fact that has contributed to the increase in global milk production, which reached 900 million tons in 2020 [4]. There is a direct relationship between population growth and the demand for dairy products [5]. Furthermore, with the rise in milk production, the volume of whey produced worldwide has also significantly increased in recent decades [6]. Alongside this, the dairy market has exhibited new consumption trends by incorporating whey into the composition and creation of new products similar to those using milk, such as milk cream, butter, and condensed milk.

Whey contains approximately 55% of the nutrients in milk: soluble proteins, lactose, vitamins, minerals, and a minimal amount of fat [6]. It is used in the food industry in both liquid and dried forms, with the dried form being categorised into concentrate, isolate, and hydrolysed forms. Table 1 illustrates this comparison between milk and whey.

Dairy production requires significant amounts of natural resources, energy for processing, and packaging materials for marketing [8]. Along these lines of reasoning, utilising whey in these products becomes an attractive option, as it involves a simple process that requires only small additional investments due to the use of existing processes and equipment [9]. Furthermore, the adoption of Industry 4.0 technologies has opened up new possibilities for the sector, streamlining the processing of products like whey [10] and resulting in fresh opportunities for companies within the industry.

The efficacy of incorporating whey into developing new foods has already been demonstrated through research [7]. Additionally, recovering valuable substances holds significant importance from the perspectives of natural resources and industrial by-products [11]. The industry has recognised this and begun to view whey as a potential ingredient, thereby establishing a new category of foods encompassing a range of newly manufactured products with a lower nutritional value than milk. The introduction of these novel products into the Brazilian market was made possible thanks to technological advancements that have rendered the processing of dairy products feasible [12].

The Brazilian Ministry of Agriculture regulates the utilisation of whey in the production of dairy products and beverages [13]. The introduction of these beverages has been presented to companies as both a technological opportunity and a means of expanding the market for whey-based products, consequently enhancing the adaptability of these dairy firms [14]. Nevertheless, companies are promoting new products derived from whey and dairy combinations and original products like whole milk, condensed milk, and milk cream. Additionally, product packaging is a widely employed strategy by these companies to attract consumer attention and drive purchases [15]. This packaging closely resembles the original product, with only minor differences, and is therefore marketed alongside the original products.

Activities within various industry sectors have been increasingly guided by concerns related to sustainability [16]. Within the dairy sector, the improper disposal of whey by the industry represents an environmentally damaging practice due to its significant polluting potential [6]. Whey’s high biochemical and chemical oxygen demand makes its disposal costly under existing environmental regulations [17]. Hence, the utilisation of whey can be connected to a growing corporate concern regarding environmental factors.

In the dairy industry, the production system must exhibit high levels of reliability, low costs, superior quality, and enhanced productivity [18]. Moreover, developing new products encompasses various stages, including conception, research, analysis, development, and launch [19]. In the pursuit of generating profits during challenging times, a recent trend has emerged towards the development of technologies that have rendered dairy product processing feasible, resulting in the creation of new whey-based beverages [12]. Additionally, the option to manufacture novel dairy products from milk by-products has proven to be a strategy for expanding product portfolios, customer bases, and market shares of dairy companies [20]. This approach enhances productivity, reduces waste generation, and allows for the reutilisation of by-products to comply with legal and regulatory requirements [21].

The attainment of competitive levels of success for companies within the dairy sector hinges on their capacity to fulfil strategic objectives related to their products and processes [22]. To achieve favourable outcomes in these strategic objectives, operations management employs a range of metrics that enable the analysis and comparison of results from internal and external processes [23]. In this context, competitive factors significantly positively influence an organisation’s competitive performance [24]. Competitive success factors are areas where a company must achieve positive outcomes to translate strategies into tangible actions and realise intended results and strategic objectives [25,26]. Consequently, an array of socioeconomic, demographic, and biological factors impact the consumption of dairy products [27]. Hence, it becomes evident that numerous competitive factors are linked to the production and utilisation of whey in the dairy industry. Given the considerations surrounding environmental, nutritional, and market-related concerns, it is apparent that there exists a research gap that warrants exploration and investigation.

Hence, this research aimed to identify and analyse the competitive factors that influence the production and utilisation of whey within the dairy products industry while establishing connections with Brazil’s economic, nutritional, and food landscape. This aim was realised through the execution of a systematic literature review, which facilitated the identification of competitive factors. Subsequently, a hierarchical business modelling technique was employed to classify these competitive factors. Leveraging this modelling approach, the Strengths and Weaknesses, Opportunities and Threats (SWOT) constituting the SWOT Matrix were correlated with the discourse concerning the present state of the dairy products industry in Brazil.

This article aims to contribute scientifically and to business practice by identifying and engaging in discussions about the competitive factors linked to the utilisation of whey within the dairy products industry. This hierarchical arrangement of competitive factors serves as an initial point of reference for future research endeavours aimed at diagnosing and quantifying the competitive performance of companies operating in this sector. Furthermore, modelling these factors equips companies with strategies to capitalise on business prospects and navigate challenges effectively.

The subsequent sections of this article are organised as follows: Section 2 expounds on the research methodologies employed; Section 3 presents and delves into the outcomes of the study; Section 4 showcases the SWOT Matrix, aligning the findings with the context of Brazil; and finally, Section 5 delivers the research’s concluding remarks.

## 2. Methods

The methodological approach of this research was structured into two distinct stages. The initial stage encompassed a systematic literature review, wherein the primary aim was to pinpoint the factors driving the dairy industry’s adoption of whey in creating dairy blends and beverages. Following this, the subsequent stage involved the analysis and discussion of the outcomes of the review to establish linkages between these findings and the prevailing circumstances in Brazil. Figure 1 provides an overview of the methodological sequence adhered to throughout this research.

Next, the stages and steps shown in the figure are detailed.

### 2.1. Stage 1—Systematic Literature Review

The objective of the systematic literature review in this research was to explore the existing literature on the subject [28], consolidating the scientific understanding regarding the competitive factors driving the utilisation of whey in the production of dairy products. This review was methodically structured into five sequential steps.

Step 1—Definition of search terms and databases

The initial step involved the establishment of search terms to be employed and the identification of databases for conducting the searches. The central research question directing the literature review was the following: what factors propel the dairy products industry to develop novel products incorporating whey into their composition?

In pursuit of this objective, diverse terms and combinations were experimented with to identify articles that could contribute to addressing the research question. These combinations were shaped using Boolean operators such as AND and OR, allowing for the creation of combinations that would aid in responding to the research question. These combinations underwent assessment across various health and industrial management databases, both on a global scale and within Brazil. Notably, the databases yielding a greater quantity and higher quality of articles included PubMed, SciELO, and Scopus.

The PubMed database was chosen because it encompasses the most relevant research in the health area. The SciELO database brings relevant Brazilian research, while the Scopus database stores articles in the health area and has a significant collection of articles in the industrial management area. In this way, with the research question and the bases chosen, it was defined that the searches would be carried out with the following terms:
➢SciELO: (leite OR milk) AND (“soro de leite” OR whey) AND (business OR gestão OR management)➢PubMed and Scopus: milk AND whey AND “dairy beverage”.
Step 2—Search filter definition

Evidence can be found in the scientific literature from 2014 onwards regarding incentives provided by the Brazilian government for developing technologies that enable the economically and technologically viable utilisation of whey [29]. In 2017, Decree 9013/2017 was enacted in Brazil to regulate Law No. 1283 of 18 December 1950 and Law No. 7889 of 23 November 1989, both of which pertain to industrial and sanitary inspection of products of animal origin [13]. In light of this, the search timeframe for databases was determined to be from 2014 to the present, encompassing the period starting with the initiation of Brazilian government incentives and the period following the formal regulation of whey utilisation in dairy product composition. Both national and international articles were researched, as whey is not only a prevailing trend in Brazil but also finds application in the composition of dairy products in other countries.

Step 3—Search in databases

During this third phase, searches were conducted across various databases, encompassing articles, review articles, and conference proceedings. In the preliminary searches, employing the terms and filters outlined in steps 1 and 2, 790 articles were identified in the PubMed database, 131 in the Scopus database, and 56 in the SciELO database. Apart from these searches, eight highly pertinent articles and documents were incorporated closely aligned with the subject matter. These supplemental resources were discovered through supplementary research and addressed at least one of the investigated factors. Including these materials enriched the review with valuable insights, complementing and augmenting the relevant information gathered.

Step 4—Reading the titles and abstracts of articles

Moving on to the fourth stage, the titles and abstracts of the articles were carefully reviewed to ascertain their alignment with the research theme. Following this assessment, 33 articles were deemed suitable and chosen for inclusion in the systematic literature review. Therefore, the combined count of documents that required examination from the databases totalled 33 articles. These were then combined with the eight articles and pertinent documents that were manually incorporated, culminating in a comprehensive set of 41 papers that constitute the content of the review.

Step 5—Complete reading of the articles

During this phase, the chosen articles were meticulously examined to identify the determinants prompting the dairy product industries to integrate whey into their product compositions. Subsequently, a structured table was created, cataloguing the identified factors. The table included references to the authors who highlighted each discerned factor and the scientific evidence they referenced. This documentation elucidated the rationale behind the presence of each specific factor.

### 2.2. Stage 2—Results Analysis and Discussion

The management models should mirror the real-world scenarios encountered [30], capturing the intricacies through a systematic structure like a variable hierarchy akin to a decision tree [31,32]. Within this structure, higher levels can be subdivided into more intricate tiers. This variable hierarchy can subsequently be integrated into decision support frameworks and subjected to assessment using multicriteria analysis methodologies [33]. Constructing a mental map proves valuable in arranging these variables, offering insight into the predicament and components of the hierarchy [34,35].

In this second phase, the identified competitive factors have been systematically organised into distinct domains encompassing nutrition, technology, economics, organisation, and the environment, thus resulting in a two-tiered hierarchical structure. This structuring facilitated the steering and organisation of discussions, fostering the connection and integration of factors that shared similar attributes within their respective domains. Moreover, armed with the wealth of sourced articles, these discussions could be further enriched and explored, drawing from pertinent scientific literature that expounds upon the origins of the identified factors.

This stage also incorporated the application of the SWOT Matrix, denoting strengths, weaknesses, opportunities, and threats. Within this framework, strengths and weaknesses pertain to internal facets of the companies [36], while opportunities and threats relate to the external environment [37]. Thus, the matrix was devised to comprehend the strengths and weaknesses associated with using whey in dairy product manufacturing. Additionally, the matrix aided in identifying potential business opportunities to be explored and the primary threats within this market landscape.

In summary, the research methodology is bifurcated into two principal stages: first, a systematic literature review, and second, an analysis and discussion of the obtained results.

## 3. Results and Discussion

Upon reviewing the 41 articles and research documents from the initial stage, 17 distinct competitive factors emerged, elucidating the impetus driving dairy product industries to incorporate whey in their product formulations. These competitive factors were identified as elements that potentially substantiate the inclusion of whey within dairy products, labelled from Factor 1 (F1) to Factor 17 (F17). Table 2 provides an overview of these factors, furnishing their descriptions, the frequency of citations, the percentage of articles referencing each factor, and corresponding references.

The research also revealed a variation in the citation count for each factor, with four of these factors being referenced by more than ten authors: F1—Nutritional composition of products improvement; F4—Reduction of the environmental impact; F2—Sensorial aspects of products improvement; and F8—New production technologies.

The identified competitive factors were systematically arranged into a hierarchical structure resembling a decision tree. This arrangement was guided by the competitive factors and their primary spheres of impact, further divided into nutritional, environmental, economic, technological, and organisational domains. Such an organisational approach was conceived to enhance comprehension, guiding discussions along trajectories drawn from the trends identified while examining literature review articles. As a result, Figure 2 provides an illustrative representation of the hierarchical alignment of competitive factors.

The following subsections will present detailed discussions of the competitive factors according to the hierarchy defined in Figure 1.

### 3.1. Nutritional Factors

Factor F1—Nutritional composition of product improvement—emerged as the most frequently cited competitive factor among the authors featured in the literature review. This observation underscores how whey has been embraced within the industry as a conduit for enhancing dairy products’ composition and nutritional value. The integration of whey into formulations aims to augment nutritional profiles and elevate protein intake [47], providing an alternative avenue for deployment within the dairy sector. Consequently, this facet has piqued the interest of numerous researchers due to its nutritional, functional, and economic potential [41], further bolstered by the substantial production volumes associated with whey [44]. A prime exemplification of this trend lies in the utilisation of whey in ice cream production, which has demonstrated its viability as a substantial replacement for milk in formulations. This alternative has exhibited remarkable nutritional significance within this genre of production [52]. This substantiates the existence of numerous nutritional prospects awaiting exploration by the dairy industry, enabling the creation of products endowed with nutritional compositions that can contribute to consumer well-being.

The presentation of products, accentuating their unique attributes such as texture, flavour, and aroma, wields power to influence consumers’ favourable or unfavourable perceptions of a particular brand or item. This underscores the paramount role that sensorial attributes play in consumer purchasing decisions. Consequently, Factor F2—Sensorial characteristics of the product improvement—emerges as the second most frequently mentioned factor in the discourse. This factor’s prominence stems from its pivotal impact on product sales by shaping consumer preferences. Whey is one of the prime ingredients frequently harnessed to enhance the texture of meat products [44] and bestow a smoother flavour and consistency to yoghurts [40]. Milk’s rich whey protein content is crucial in refining texture, endowing products with heightened firmness and viscosity [47,55].

Additionally, the tendency for beverages incorporating whey to receive lower preference often relates to their relatively lower viscosity [42]. Whey contributes substantially to the sensory attributes of products it is incorporated into. The dairy industry has diligently explored this facet, capitalising on its potential to elevate product acceptability and, consequently, drive commercialisation.

Factor F3—Consumers’ health concerns—underscores the discernible shift in consumer preferences towards products that align with their health aspirations [59]. The increasing emphasis on consumer health and well-being has spurred the demand for dairy beverages containing whey as part of their composition [48]. It is worth noting that products featuring whey are not only popular among the general population, but they are also consumed by various demographic segments, including the elderly, children, and adults, given that whey proteins harbour bioactive constituents that contribute to disease prevention when integrated into the production of functional foods [44]. This growing trend of incorporating whey into dairy products is a testament to the mounting consciousness regarding the overall health and wellness of the populace.

Another crucial nutritional aspect, including whey in products, tends to enhance consumer acceptance (Factor F7). This stems from the perception that incorporating whey can render dairy beverages healthier by reducing fat content, a shift that aligns with consumer preferences [41]. Concurrently, the global consumption of milk and dairy beverages has exhibited a consistent rise (Factor F13—The growing trend in the consumption of milk and dairy products) [40], encompassing items enriched with whey-derived constituents [69]. This trajectory has propelled these ingredients into notable development, primarily driven by the burgeoning demand for functional foods, dietary products, and infant formulas [68]. It is worth noting that the surging demand for more nutritious dairy goods has engendered a necessity for extensive research on whey aimed at uncovering novel constituents and crafting premium-quality products [2]. This underscores the ongoing need for adaptations within the production sector to effectively cater to the market’s competitive and self-sufficient demands [66]. Drawing from this, it is conceivable that consumer acceptability and the escalating trend in dairy product consumption can be explored in tandem with the industry. These factors can be regarded as mutually reinforcing relationships, wherein positive advancements in one factor can potentially cascade into positive outcomes in the other.

### 3.2. Environmental Factors

The literature review unveiled two pertinent environmental factors. The initial one, Factor F4—Reduction of the environmental impact—substantiates the rationale behind leveraging whey to produce dairy beverages. The central motivation stems from the significant environmental challenge posed by disposing of unused whey, attributed to its substantial organic composition [51]. Notably, industrial effluents from using whey also exhibit analogous organic composition, albeit with a limited environmental impact due to their diluted nature [63]. Consequently, formulating fermented milk beverages employing whey sidesteps the wastage of a protein- and lactose-rich dairy product [43]. This practice concurrently curtails disposal-related costs borne by dairy companies, enhancing their competitive edge [66]. Furthermore, such an approach offers a strategic means of alleviating the environmental predicament resulting from effluent disposal [41].

The second pertinent environmental factor that underscores the utility of whey in dairy product compositions is F11—Reduction of water and energy consumption in the process. This is attributable to the lower energy demand associated with processing skimmed milk or whey, which enhances cost-effectiveness [67]. An illustrative instance of this phenomenon is using acid whey in the formulation of fermented milk products. This practice is technically viable and contributes favourably to curtailing water consumption during production [42]. The industry’s incorporation of whey as a food ingredient yields both environmental and financial benefits. It has a positive ecological impact by mitigating waste, subsequently translating into cost savings for companies that no longer need to allocate resources for proper whey disposal. Additionally, this incorporation facilitates water and energy conservation during production processes. Consequently, these saved resources can be channelled toward other initiatives, such as research and development endeavours to craft novel products and refine the company’s cost–benefit dynamics.

### 3.3. Economic Factors

Viewing the application of whey from an economic standpoint, the foremost factor to consider is F5—Reduction of production costs. Producing whole whey beverages offers reduced costs and an encouraging potential for whey valuation [50]. Decisions about the inclusion of whey protein products as ingredients revolve around the balance between the cost of the ingredient and the value it imparts in terms of functionality [39]. This cost reduction encompasses not only the curtailment of resource consumption, such as energy and water, but also hinges on the prospect of enhanced value and functional attributes inherent to whey when integrated into products. Moreover, the concentration on incorporating whey into product compositions manifests the prospect of value addition, as whey becomes a significant component, diminishing the need for other raw materials. This confluence of factors drives a reduction in overall production expenses.

F6—Product portfolio increase—also falls within the economic perspective, as it entails generating new products from by-products, proving to be an effective strategy for enriching the range of dairy offerings [20]. Additionally, the whey industry has evolved, transitioning from an environmental challenge to a realm fostering innovation by diversifying products and enhancing their functional and nutritional attributes [68,69]. Expanding the product portfolio necessitates relatively modest investments when juxtaposed with developing other dairy products, emblematic of Factor F15—Investment needs. Bernardi [46] shed light on this notion by elucidating that manufacturing dairy beverages incorporating whey utilises pre-existing equipment, demanding minor supplementary investments. The notion of broadening the product array might seem straightforward when envisaging the integration of whey into various novel products. However, this factor is intrinsically intertwined with the requirement to introduce products that deliver substantive nutritional value to consumers rather than merely serving as conduits for generating profit through marketing low-value, nutritionally deficient items. This factor shares a connection with investment needs. While indications might point towards a relatively modest investment demand, it remains imperative that these investments align with the genuine necessity to fabricate high-quality, value-enhanced products with favourable outcomes for consumers.

Another factor the dairy industry has explored is the increase in customers and the expansion of market share (F16) [20]. This phenomenon propels an escalation in the assortment of available products, effectively broadening the scope to cater to diverse consumer segments and consequently enlarging companies’ market shares. This drive to expand consumer reach correlates with an economic perspective, emphasising growth and competitive positioning.

Furthermore, Factor F17—Trade dress and the use of similar packaging—is another compelling competitive factor rooted in an economic context. This factor operates on the principle of leveraging the packaging of well-established products within a particular product line, leveraging their recognition and credibility. By infusing similar visual identities into related products, companies can harness the established image and promotion of a leading product to bolster the identity of comparable items with minimal discrepancies [74]. This strategy holds favourable implications for lesser-known enterprises seeking entry and competition within the market. However, it is noteworthy that similar packaging can also be employed to market lower-quality alternatives intended to supplant higher-quality counterparts. As instances of this approach, consider the marketing of whey-based dairy drinks as substitutes for whole milk and the promotion of condensed milk blends as alternatives to traditional condensed milk.

### 3.4. Technological Factors

Shifting the focus to technological dimensions, the forefront factor to consider is F8—New production technologies. In this context, the utilisation of whey ushers in a realm of innovation by fostering the creation of new products characterised by enhanced nutritional and sensory attributes, opening avenues for implementing pioneering processing techniques [59]. This realm of innovation permits diversifying products manufactured on a single production line, thereby augmenting the capacity of these lines. An instance is the employment of technologies devised for recuperating whey components boasting elevated nutritional and technological profiles, which can be converted into value-added products [2]. Another illustration of these novel technologies involves harnessing whey sourced from butter and cheese to enhance ice cream production, heralding a high-tech, high-nutrition, environmentally conscious, and socioeconomically impactful alternative [52]. Further exemplifying this is the ultrafiltration of whey protein and lactose [64]. These instances underscore the emergence of enhanced and economically viable separation and purification technologies for whey proteins, alongside the continued exploration of new sources. This trend is poised to perpetuate the industry’s growth [69]. Consequently, these pioneering production technologies enhance quality and efficiency within production processes and present fresh avenues to capitalise on whey as a foundational component for crafting diverse food varieties.

Another technological factor, F14—Physicochemical properties of product improvement—has justified the application of whey by the dairy industry. Isolated whey proteins also find applications within the dairy industry due to their physicochemical and nutritional properties [45]. Additionally, the use of whey enhances the physicochemical properties of the products. This indirectly contributes to enhancing the sensory characteristics of the products, making them more appealing to consumers.

### 3.5. Organisational Factors

In the literature review, three organisational factors were also highlighted. These factors are aspects that influence companies in matters of process management. The first factor in this regard was F9—Optimisation of production processes. This optimisation and improvement of milk and cheese production processes aim to recover value-added by-products [51], which are efficiently used as raw materials for other products and allow the utilisation of secondary industrial processes [71]. Moreover, several techniques have been developed for food formulations using whey protein aggregates. However, limitations still exist regarding their industrial applications due to the water retention capacity of denatured whey proteins [40]. Additionally, the requirement for multiple process streams for whey underscores the importance of process optimisation for each type of dairy stream [68]. Therefore, process optimisation pertains not only to enhancements in each production flow of dairy products but also to adapting production flows to enable the production of more than one product in each flow.

The second organisational factor identified was F10—Adding value to the supply chain. The utilisation of whey in dairy product production also aims to enhance the overall value of the production chain [52]. In this context, this practice presents a valuable opportunity to enhance the competitiveness of the entire dairy chain. This involves creating new product pathways among agroindustries, reducing waste, cutting costs, and generating employment opportunities and fresh sources of income within the chain [66]. The third related organisational factor is F12—Adding value to products. Thanks to its rich nutritional composition, whey can be incorporated into other dairy products to augment their value [73]. This transforms whey from a mere by-product into a valuable raw material for producing higher value-added items like probiotic beverages [57]. It can even be processed into a resource for widespread usage across the food, chemical, and pharmaceutical industries [66]. Adding value to the supply chain and products brings substantial advantages to the dairy industry. It underscores organisations’ need to establish a coherent strategic plan to leverage the potential of whey utilisation fully.

This section presents a hierarchical model and discussion of the 17 competitive factors identified, considering perspectives from nutrition, environment, economics, technology, and organisation.

## 4. SWOT Matrix

Through the identification and comprehensive discussion of the competitive factors driving the incorporation of whey into dairy product compositions, a range of strengths, weaknesses, opportunities, and threats associated with this market trend have emerged for companies aiming to establish themselves in this sector. Figure 3 illustrates the organisation of these strengths, weaknesses, opportunities, and threats within a SWOT Matrix.

Bringing the discussion to the context of Brazil, we can highlight the growing market and technological development as strengths. This can be substantiated by the fact that Brazilians consume an average of 51 L per year of milk [75]. Moreover, the milk production industry in Brazil has undergone a modernisation process, marked by the introduction of disposable packaging and the widespread popularity of dairy products like yoghurt and dairy desserts [76]. A noteworthy advantage is the health benefits associated with the increased consumption of dairy products in Brazil. This trend has been primarily influenced by the ageing population and evolving consumer habits, which consistently seek healthy and distinctive food options [77].

The utilisation of whey in various products along the dairy chain is regulated in Brazil by the Ministry of Agriculture [13]. In conjunction with this regulatory framework, the Brazilian dairy industry recognises consumer acceptance as a promising opportunity. This has led the industry to focus on developing functional foods for disease prevention. Consequently, the dairy sector has started producing beverages with reduced fat content and enriched with functional ingredients [78]. Incorporating whey into the composition of dairy products presents an avenue for expanding the customer base. This expansion is facilitated by the product portfolio’s diversification while yielding positive effects on the value chain. What was previously regarded as a by-product now assumes the role of a valuable raw material for creating other products. This transition has beneficial ripple effects on the entire value chain.

Conversely, the sector faces weaknesses. One such weakness is the incomplete reutilization of all generated whey, shedding light on the issue of environmental pollution if proper disposal practices are not adhered to. Another drawback is the production of subpar products in Brazil that employ whey as their primary ingredient. One example is a milk drink with approximately 60% whey content in its composition. This product has been found in consumption, particularly among Brazilian households receiving federal government assistance, where 31% have swapped milk for whey [79]. This milk substitution for other products has increased health problems within the broader population [80].

The introduction of new products can indeed be perceived as a threat. The development and marketing of items like whey milk drinks and condensed whey milk mixes carry the potential to have a negative nutritional impact on the low-income population. These products are being consumed as substitutes for traditional milk and condensed milk. This substitution pattern raises concerns about potential dietary imbalances among individuals stemming from the absence of crucial nutritional components essential for a balanced diet [81]. Elements like lipids, carbohydrates, and proteins, which are mandatory for consumption, may be lacking due to this dietary shift.

An additional threat pertains to the utilisation of similar packaging for distinct products. For instance, consider the case of condensed milk mixtures and powdered milk compounds. In this scenario, companies exploit the trade dress to promote these items with a higher profit margin. This strategy capitalises on their reduced production costs while selling them at prices that closely resemble the rates of the original products, namely condensed milk and powdered milk. This practice could potentially mislead consumers and compromise their purchasing decisions, ultimately affecting their satisfaction and value perception [74].

The weaknesses and threats that have been discussed indeed present significant challenges for the dairy products industry that incorporates whey into its compositions. Nonetheless, these challenges can be mitigated by effectively harnessing the strengths and opportunities outlined in the SWOT Matrix. Doing so can yield substantial benefits for the population, including providing quality products at reasonable prices and positively impacting public health.

This section has provided an overview of the SWOT Matrix and its relevance within the context of Brazil. It illustrates how the competitive factors within the Brazilian industry influence the utilisation of whey in various dairy products. Through strategic utilisation of strengths and opportunities, the industry can effectively address weaknesses and counteract potential threats, ultimately contributing to a more positive outcome for both the industry and the consumers.

## 5. Conclusions

This research successfully identified and examined 17 competitive factors that influence the production and utilisation of whey in the dairy industry. These factors were systematically categorised and analysed through nutrition, environment, economics, technology, and organisation lenses. Building upon this comprehensive analysis, strengths, weaknesses, opportunities, and threats pertinent to this productive sector were enumerated and thoroughly discussed. Importantly, these discussions considered the unique context of the Brazilian economic, food, and nutritional landscape.

This research holds academic implications as it compiles and hierarchically models the factors of competitiveness that influence organisational performance in the production and utilisation of whey within the dairy industry. Through this, new research can be conducted to apply this model and assess the competitive performance of the sector, offering insights into strategies that can be implemented to adjust the production processes within the sector.

This research also carries practical implications and can aid in developing nutritional strategies for creating new products that incorporate whey into their composition, offering consumers convenience, practicality, and nutritional quality. Furthermore, companies within the sector will be able to adopt benchmarking practices to enhance and standardise their processes in response to identified competitive factors. They will also strive to implement evaluation tools based on indicators to monitor the achievement of results associated with each competitive factor. Lastly, the research provides evidence from a social standpoint, demonstrating that individuals with lower incomes are more inclined to consume whey-based products as substitutes for milk due to their lower cost, potentially leading to adverse health consequences.

Limitations were identified within the scientific literature and other published works, as these materials briefly touched upon the theme without delving into in-depth discussions. For future research, it is recommended to conduct a survey involving nutrition professionals and managers within the dairy product industry. This survey would assess the influence and significance of the factors described in this research on the development of products containing whey in their formulation and the overall management of companies operating within this food sector.

## Figures and Tables

**Figure 1 foods-12-03348-f001:**
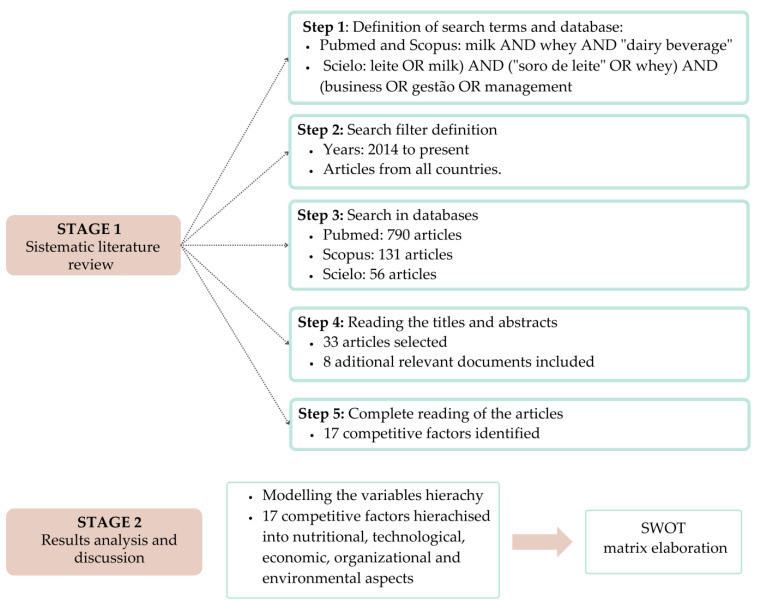
Methodological flow of the research.

**Figure 2 foods-12-03348-f002:**
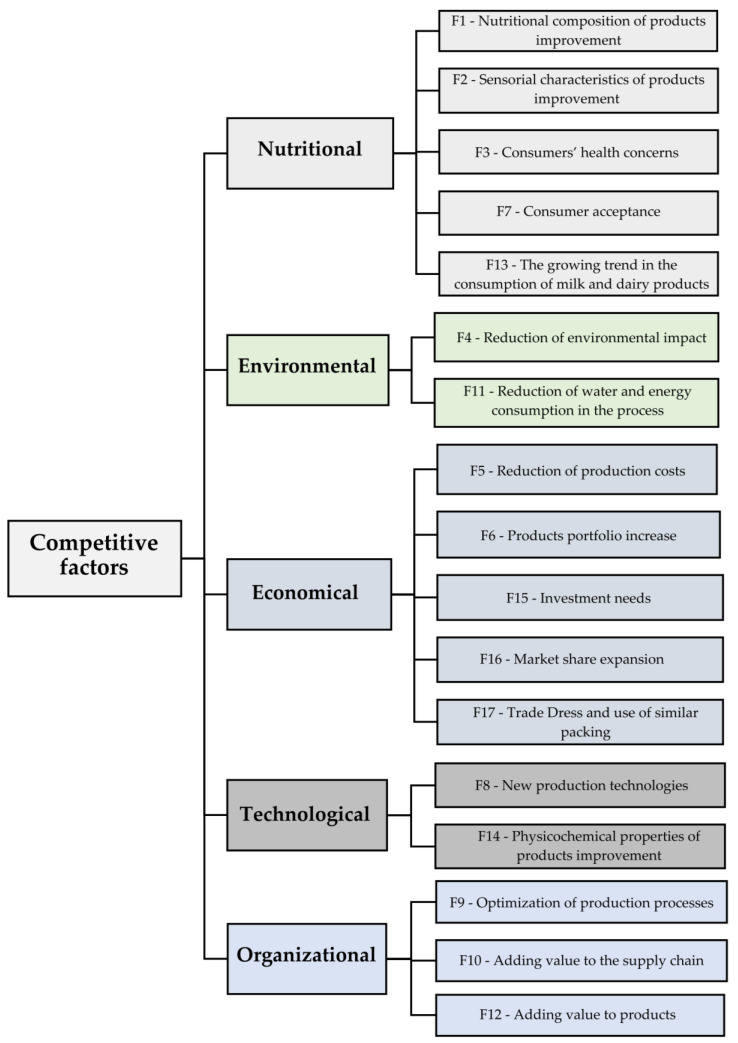
Hierarchical modelling of competitive factors related to the use of whey in the dairy industry.

**Figure 3 foods-12-03348-f003:**
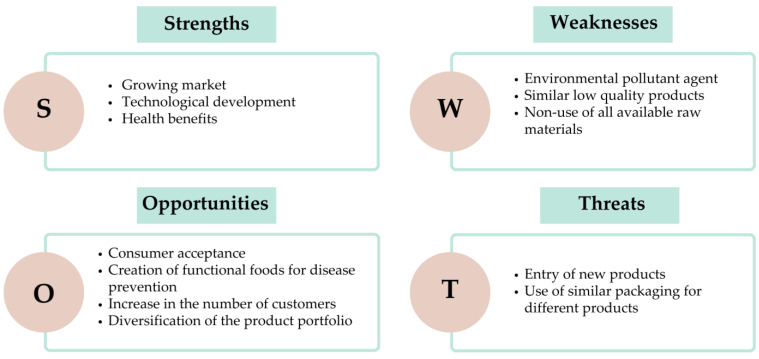
SWOT Matrix for market trends on the use of whey by the dairy industry.

**Table 1 foods-12-03348-t001:** Nutritional comparison between milk and whey (per 100 mL).

Constituent	Milk	Whey
Lactose (g)	5.25	4.6
Lipid (g)	3	0.5
Protein (g)	2.9	0.6
Calcium (mg)	120	47
Sodium (mg)	44	54
Potassium (mg)	179	160
Magnesium (mg)	11	8
Phosphor (mg)	95	40

Source: Adapted from Oliveira et al. [7].

**Table 2 foods-12-03348-t002:** Competitive factors for the use of whey in the dairy industry.

	Competitive Factor	Number of Citations	Percentage	References
F1	Nutritional composition of product improvement	19	46.3%	[2,6,20,38,39,40,41,42,43,44,45,46,47,48,49,50,51,52,53]
F2	Sensorial characteristics of product improvement	16	39.0%	[2,20,39,40,41,42,44,47,48,54,55,56,57,58]
F3	Consumers’ health concerns	14	34.1%	[44,46,48,59,60,61]
F4	Reduction of environmental impact	12	29.3%	[38,39,40,41,42,43,46,49,51,52,61,62,63,64,65,66]
F5	Reduction of production costs	9	22.0%	[39,41,50,52,54,58,62,66,67]
F6	Product portfolio increase	8	19.5%	[6,20,45,68,69]
F7	Consumer acceptance	6	14.6%	[41,50]
F8	New production technologies	6	14.6%	[2,46,51,52,57,64,65,66,68,69,70]
F9	Optimisation of production processes	6	14.6%	[38,40,42,51,65,67,68,71,72]
F10	Adding value to the supply chain	5	12.2%	[52,66]
F11	Reduction of water and energy consumption in the process	4	9.8%	[42,53,67,68]
F12	Adding value to products	2	4.9%	[42,44,46,57,66,73]
F13	The growing trend in the consumption of milk and dairy products	2	4.9%	[2,40,44,66,68,69]
F14	Physicochemical properties of product improvement	2	4.9%	[45]
F15	Investment needs	1	2.4%	[9,66]
F16	Market-share expansion	1	2.4%	[20]
F17	Trade dress and use of similar packaging	1	2.4%	[74]

## Data Availability

Data will be made available on request.

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
