# Peer review of "A Literature Database Review of the Competitive Factors That Influence the Production and Use of Whey in the Brazilian Dairy Industry"

_foods, 2023, doi:10.3390/foods12183348_

Round 1
Reviewer 1 Report
Dear Authors,
Congratulations for the sound work you have done. My suggestions for the improvement of the manuscript are listed below
1. Table 1 and Table 2 could be merged into 1 table with 4 columns: Competitive factors, Number of citations, Percentage and References
2. Line 218: A suggestion in regards to this: the whey by itself does not enjoy widespread consumer acceptance. Therefore, endeavors are being undertaken to create beverages and other whey products that are enriched with ingredients to enhance the appeal of whey to consumers. If there is a reference within those included in the manuscript with regard to the aforementioned, it would be interesting to include it.
3. Line 335: Please consider the term “consumer acceptance” (as in the previous sections) instead of “consumer behavior” in the SWAT Matrix (Figure 2) and discussion.
4. Line 344: “Another weak point is the production of low-quality products, such as whey, sold as a milk drink containing around 60% whey in its composition“ – this sentence should be rephrased, because whey is not a low-quality product, only it is lower in some nutritional aspects compared to milk. Please make it clear.
5. Line 347: The assertion that substituting milk with whey or any other products has led to an increase in obesity and dyslipidemia appears to be rather challenging and speculative. It would significantly enhance the credibility of this claim if there were a comprehensive and substantiated explanation provided. Alternatively, presenting the claim with a more cautious tone could better reflect the complexity of the issue.
6. Line 352: This could potentially generate food imbalance..
Author Response
Thank you for your review efforts on this article. Please check the attached file for explanations about the changes and authors' responses.

Reviewer 2 Report
The manuscript “Analysis of the Competitive Factors that Influence the Production and Use of Whey in the Dairy Product Industry” is a review focused on the factors that influence the production and use of whey in the dairy products industry, with reference to the Brazilian economic, food, and nutritional reality.
The topic of the review is interesting, even if the Whey in the Dairy Product Industry is not a very novel topic. In the context of the circular economy, finding ways to use by-products to formulate food with added value is of interest.
1. The Results and Discussion section presents as a weak point the lack of a critical discussion of the information. The major findings of the analyzed article are presented without a critical intervention of the review’s authors. It is more of a string of the results of other articles on this topic.
2. The purpose of the work is not sufficiently justified, please list other review articles in which a similar subject was reviewed and explain what is original and new in the current article.
3. Methodological procedures is right term or Methods (line73)
4. Please improve table 1 format and alignments.
5. Moreover, the aim of the paper as it was stated (relating them to the Brazilian economic, food, and nutritional reality) by the authors was not demonstrated well, only in one paragraph (line 327-333) it was discussed.
6. Please discuss the timeframe of publication you have used for analysis of the topic.
7. At the end of each subchapter, it is worth providing a short but specific summary.
Author Response

(The authors gave the same response as above.)

Reviewer 3 Report
The work deals with an important research topic. The use of whey, by-products in the dairy industry, is a very current topic, important both for the industry and the average consumer. The evaluated work is a valuable study, presenting the main factors in favor of the use of whey (nutritional, environmental, economical, technological, organizational), however, the individual factors are discussed in very general terms. This is a review work, so the topic should be presented exhaustively. In addition, the work requires additions, there are missing e.g. information: about the type of whey in question, about its health benefits (it would be worth presenting the composition of whey in the form of a table), the form in which it is used in the food industry (liquid, dried, concentrates, isolates, etc.).
Author Response

(The authors gave the same response as above.)

Reviewer 4 Report
The manuscript entitled “Analysis of the Competitive Factors that Influence the Produc tion and Use of Whey in the Dairy Product Industry.” Is about the factors that influence the production and use of whey in the dairy products industry, relating them to the Brazilian economic, food, and nutritional reality. The given information is in this manuscript is useful but it is area specific and not covering the global aspects. Furthermore, the different section and required related citation to improve theover all quality of the manuscript.
· Title: reconsider as per the review objectives - Title is not indicating it as a review
· Abstract –rewrite-poorly structured
· Introduction-this is too short and required related latest citation –
· L-53-cross check with latest data- Despite its use by the industry, about half of the whey generated has been discarded 52 incorrectly, constituting a practice that is harmful to the environment due to the great 53 polluting potential [3
· L-75Creating confusion- This research can be methodologically characterised as an Observational Exploratory 74 Bibliographic Study. The exploratory study seeks to deepen research on a topic or area 75 with little information available [19], providing greater familiarity with the problem, mak- 76 ing it more explicit, and improving ideas [20]. The observational method was applied to 77 this research, seeking to observe information’
· Need clarity- The increase in the number of customers and the market share expansion (F16) is 260 another opportunity that is being explored by the industry with the use of whey in the 261 composition of a growing number of product
· Cite latest references- The variation in potential for gas retention among wheat flour doughs was mainly attributed to differences in the large-deformation properties of dough films (van Vliet et al. 1992).
· Cite the following latest references.
· Klapkiv, J., Putsenteilo, P., & Karpenko, V. (2023). The Convergence of Factors That Affect the Dairy Product Market: A Comparative Analysis of European Union Countries. Comparative Economic Research. Central and Eastern Europe, 26(2), 105-127.
· Rosa, E., & Prudencio, E. S. (2023). A Comprehensive Approach About Comparison Between Drying Technologies And Powdered Dairy Products. Food Research International, 113326.
· Please avoid repetition-
· Please check reference style throughout MS
· Italic all the scientific names,
· Remove grammatical mistakes
· Need to rewrite the conclusion
· Recheck Legends description is as per figure number and discussion-
· I urge the authors to improve the English language for better flow of literature
I urge the authors to improve the English language for better flow of literature
Author Response

(The authors gave the same response as above.)

Reviewer 5 Report
-Somehow, nice and new approach of manuscripts/work on the use of whey in the life, R&D and technology/food-dairy industries... which can be interesting to readers, but it is unclear, lacks experimentations and modeling, and should be harshly revised and improved for strength and clarity. It needs to be addressed some modeling notions and figures. The version at its present form lacks such vigor and the authors should work hard to make this nice topic more reader friendly and useful.
-title is misleading. Should be improved in a professional manner.
-When I saw the title, I immediately thought waoooo it would be interesting work... but when you read the entire text in is not so.
-The competitive factors should be addressed with using some model to better relate them.
-The main issue here is very difficult to follow.
-In the captions of the tables and figures please very clearly add what is the main message of each?
-Some minor issues related to the fonts and (semi)colon and points throughout the text should be precisely applied.
-
Author Response

(The authors gave the same response as above.)

Round 2
Reviewer 2 Report
In its current form, the manuscript is suitable for publication.
Author Response
Answers to Reviewer R#2:
General Comment – R#2: In its current form, the manuscript is suitable for publication.
Answer to General Comment – R#2: Thanks for reviewing the article.
Reviewer 3 Report
I accept the response to my comments.
Author Response
Answers to Reviewer #3:
General Comment – R#3: I accept the response to my comments.
Answer to General Comment – R#3: Thanks for reviewing the article.
Reviewer 4 Report
The authors have adequately responded to the raised comments and the revised version of the manuscript is acceptable for publication after the following minor revision-
-Please recheck factor F1 with respect to the hierarchy Factor F1 – Nutritional composition of product improvement was the competitive 234 factor most cited by the authors, demonstrating that there is relevance on the part of the 235 industry in improving the nutritional composition of products through whey How this idea is viable for industry- In this sense, using whey provides opportunities for inno- 350 vacations in product development and the implementation of new processing technologiesMinor editing of English language required
Author Response
Please check our responses to the recommendations in the attached file.

Reviewer 5 Report
This somehow different style of nice study and focus still needs revision, improvement and corrections in order to be more user friendly fig(s)…
Because this review is based on database, I would add the word databases in the title. Please professionally elaborate.
-A schematic reader friendly fig for the methodology of this unusually zero-experimental work need.
-Some better words and term should be used throughout… for example, … “the utilisation of whey”… etc.. it should be replaced by other more appropriate term like (use, application etc…). please elaborate throughout the text.
-Table 1. Nutritional…., for this caption please (on the caption) add from which references (more appropriate analytical ones)? Also how about its Nan Cl and other ions contents? please add.
-Lines 110-111, ….the strengths and weaknesses, opportunities and threats that make up the Strengths, Weaknesses, Opportunities, and Threats (SWOT)…. Please write this in a more professional manner with no repetition. Readers might not like.
-Line 155, Mentions can be found…… “Mentions”?
-English should harshly be revised and improved for clarity and simplicity (with more professional words, sentences).
-more precise use of fonts and punctuations throughout the text.
Good luck
-
Author Response

(The authors gave the same response as above.)
